# Deciphering Invariant Feature Decoupling in Source-free Time Series Forecasting with Proxy Denoising

## Abstract

The proliferation of mobile devices generates a massive volume of time series across various domains, where effective time series forecasting enables a variety of real-world applications. This study focuses on a new problem of source-free domain adaptation for time series forecasting. It aims to adapt a pretrained model from sufficient source time series to the sparse target time series domain without access to the source data, embracing data protection regulations. To achieve this, we propose TimePD, the first source-free time series forecasting framework with proxy denoising, where large language models (LLMs) are employed to benefit from their generalization capabilities. Specifically, TimePD consists of three key components: (1) dual-branch invariant disentangled feature learning that enforces representation- and gradient-wise invariance by means of season-trend decomposition; (2) lightweight, parameter-free proxy denoising that dynamically calibrates systematic biases of LLMs; and (3) knowledge distillation that bidirectionally aligns the denoised prediction and the original target prediction. Extensive experiments on real-world datasets offer insight into the effectiveness of the proposed TimePD, outperforming SOTA baselines by 9.3% on average[1].

## 1 Introduction

The widespread deployment of Internet-of-Things (IoT) sensors has produced massive time series data across domains (Sun et al., 2025; Wang et al., 2024a), including traffic (Kieu et al., 2024; Cirstea et al., 2022), weather (Hettige et al., 2024), and energy (Wu et al., 2020). Accurate time series forecasting is crucial, enabling effective decision-making across diverse domains (Liu et al., 2025a; 2024a; Campos et al., 2023). We are seeing impressive advances in machine learning, especially in deep learning, that are successful in effective feature extraction and value creation (Hettige et al., 2024; Liu et al., 2025b). They are mainly dedicated to creating models based on large amounts of domain-specific time series (see Figure 1(a)). However, in real-world scenarios, the time series data can be sparse due to various reasons, such as data collection mechanisms (e.g., low sampling rate) and data privacy. The performance of existing time series forecasting methods may degrade remarkably with such insufficient training data (Jin et al., 2022).

Although recent research efforts have been devoted to addressing sparse training data by means of transfer learning (Pan & Yang, 2009; Wang et al., 2024a), these are mainly designed for computer vision and natural language processing while ignoring the specific characteristics of time series, i.e., capturing complex temporal correlations (Shao et al., 2025). In addition, these methods are often performed across domains by leveraging both source and target data (Liu et al., 2024b). However, the reliance on source data may raise various concerns, e.g., training inefficiency and data privacy. Further, large language models (LLM) based methods emerge as a new paradigm for universal time series forecasting. Nonetheless, it is expensive to train these large models, incurring high computational costs. Further, despite LLM-based methods offering acceptable time series forecasting performance, they often fail to achieve superior performance on specific domains, especially domains with scarce time series. To address these issues, this study focuses on a new problem of source-free domain adaptation (Tang et al., 2025; Ragab et al., 2023) for time series forecasting, referred to as

---

[1]The code can be found at `https://anonymous.4open.science/r/TimePD-52E7/`.

Figure 1: (a) Domain-specific time series forecasting. (b) Source-free time series forecasting. (c) Limited target data acquisition. (d) Cross-domain distribution shift.

source-free time series forecasting (SF-TSF), as shown in Figure 1 (b). The SF-TSF aims to directly adapt a source model to a target domain using only its parameters, without accessing its source data.

However, it is non-trivial to develop SF-TSF methods due to the following challenges. First, in the source-free domain adaptation scenarios, it is often hard to acquire sufficient target data due to privacy and data collection mechanisms. Limited target data acquisition often results in insufficient observations, as shown in Figure 1 (c). It is challenging to effectively capture the complex temporal correlations of such sparse target data. Second, alleviating the cross-domain distribution shift between source and target data also poses difficulties. Owing to differences in data sensing mechanisms, the statistical properties of time series, such as trend and season, of the target domain may largely deviate from those of the source domain. The distribution shift makes it difficult for most existing time series modeling methods trained on the source domain to generalize to the target domain, as shown in Figure 1 (d), resulting in low effectiveness. Recent advances have incorporated LLMs into time series modeling, benefiting from their pre-trained knowledge and generalization ability (Huang et al., 2024). However, this naturally introduces the third challenge: how to effectively leverage the knowledge embedded in LLMs while alleviating noise, since LLMs are prone to hallucinations (Sriramanan et al., 2024), producing irrelevant or misleading outputs when faced with domain-scarce signals, which could distort the forecasting.

This study addresses the above challenges by providing a novel Source-Free **Time** Series Forecasting framework with **P**roxy **D**enoising (TimePD). To facilitate effective temporal correlation extraction across sparse target time series, we develop an innovative source-free domain adaptation paradigm, where the target model borrows the rich knowledge learned on sufficient source data based on the assumption that time series from different domains share certain latent patterns (Jin et al., 2022). Further, we achieve completely invariant disentangled feature learning, which also alleviates cross-domain distribution shift. Specifically, we design a dual-branch architecture that explicitly decomposes input series into seasonal and trend components and enforces invariance at both the representation and gradient levels. Stochastic augmentation and specialized invariance blocks further strip away component-specific cues, obtaining disentangled and component-invariant representations. To leverage the transferability ability of LLMs while alleviating hallucinations, we applied pre-trained LLMs to guide the target model, alleviating the impact of domain shift on the target model. Then, we introduce a proxy denoising mechanism, which treats LLM as a powerful but probably noisy proxy forecaster, to denoise the LLM's forecasts. It dynamically corrects its systematic bias on the target domain by leveraging the consensus between the source model and the adapting target model, producing more reliable forecasts for subsequent guidance. Then, we establish a bidirectional knowledge transfer loop: denoised proxy forecasts supervise the target model, while target predictions feed back to stabilize the proxy correction, preventing distribution drift from the target domain. Finally, we employ knowledge distillation to further calibrate the target prediction with the denoised prediction, enhancing model performance.

The main contributions are summarized as follows:

- To the best of our knowledge, this is the first study to learn source-free time series forecasting and propose an LLM-empowered framework called TimePD that unleashes the power of LLMs and models trained on sufficient data to improve.

- We propose an invariant disentangled feature learning method to handle the cross-domain distribution shift, a proxy denoising strategy to alleviate the hallucinations of LLMs, and a knowledge distillation mechanism to transfer denoised knowledge from LLMs to a lightweight target model.

- Extensive experiments on real-world datasets demonstrate that TimePD outperforms state-of-the-art baselines, achieving average improvements of **10.7%** and **9.3%** in terms of MSE and MAE, respectively, offering a brand new paradigm for cross-domain time series analytics.

## 2 RELATED WORK

**Time Series Forecasting.** Time series forecasting attracts increasing interest due to the growing availability of time series data and rich downstream applications (Benidis et al., 2022; Hettige et al., 2024). Traditional time series forecasting models (Box et al., 2015) are mostly based on shallow statistics, making them difficult to capture the complex temporal correlations. Recent advances in deep learning techniques apply neural networks for effective time series modeling (Zhou et al., 2022; Bai et al., 2020), including various architectures, e.g., CNNs (Wu et al., 2023), RNNs (Lai et al., 2018), and Transformers (Wu et al., 2021; Liu et al., 2024c). LLM-based methods (Liu et al., 2025b; Zhou et al., 2023; Jin et al., 2024) emerge as a new paradigm for time series forecasting empowered by their powerful general feature extraction capabilities. However, most existing methods require sufficient training data. Their performance may degrade remarkably when training on sparse data.

**Source-free Domain Adaptation (SFDA).** SFDA aims to adapt pre-trained models to target domains without accessing source data (Kundu et al., 2020; Kim et al., 2021; Fang et al., 2024; Mitsuzumi et al., 2024; Zhang et al., 2024). For example, SHOT (Liang et al., 2020) leverages information maximization and self-supervised pseudo-labeling. NRC (Yang et al., 2021) introduces neighborhood clustering to improve adaptation stability. However, these methods are primarily tailored for computer vision and natural language processing (Li et al., 2024), and cannot capture the unique temporal correlations among time series. Although recent studies (Ragab et al., 2023; Zhong et al., 2025) have explored SFDA for time series imputation, its application to forecasting remains largely underexplored. At the same time, large language models (LLMs) have demonstrated the ability to acquire generalized knowledge across diverse tasks, showing strong potential for time series forecasting (Jin et al., 2024). However, most existing SFDA approaches have yet to effectively harness the knowledge embedded in LLMs.

## 3 METHODOLOGY

Figure 2 presents the overview of TimePD, which integrates an invariant feature disentanglement learning module, a proxy denoising module, and a knowledge distillation. Sequentially, TimePD begins with training a source model $\theta_s$ on source data. Without revisiting the source dataset, $\theta_s$ is copied to initialize the target model $\theta_t$, which adapts to the target domain. Meanwhile, a pre-trained LLM $\theta_{ts}$ is applied for extracting knowledgeable features, which are further calibrated by the proxy denoising module to alleviate hallucinations. Finally, the knowledge distillation module is designed to minimize disagreement between the corrected proxy forecasts and target predictions.

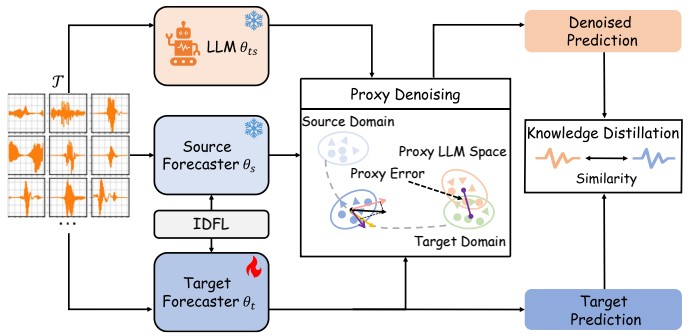

Figure 2: Overview of TimePD. Invariant Disentangled Features Learning (IDFL) is designed to boost forecasters to learn invariant features by disentangling the seasonal and trend components. Proxy Denoising aims to denoise the LLM's outputs.

### 3.1 INVARIANT DISENTANGLED FEATURE LEARNING

Invariant Disentangled Feature Learning (IDFL) aims to handle cross-domain distribution shift. As shown in Figure 3, IDFL consists of a decomposition block, forecasters, a representation-invariant block, a gradient-invariant block, and a Fourier transform module. It decomposes the input series into trend and seasonal components and learns component-invariant representations. The invariance features remain stable while other factors change (Parascandolo et al., 2021). For example,

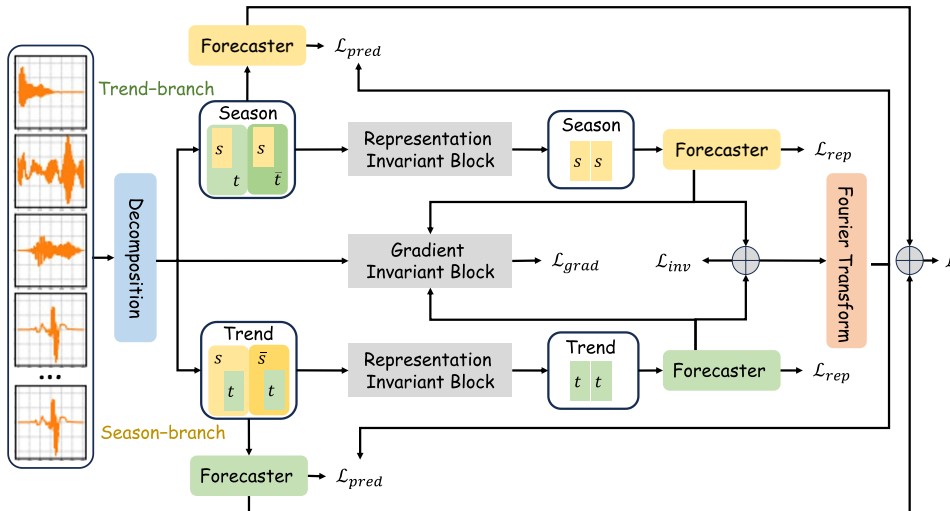

Figure 3: Model Training via **I**nvariant **D**isentangled **F**eature **L**earning (IDFL).

trend features should stay consistent under seasonal variations, and vice versa. Such disentangled invariants enhance forecasting accuracy across domains. We design two complementary branches: the *trend branch*, where seasonal variations act as domains; and the *seasonal branch*, where trend variations act as domains. To improve generalization, invariance is explicitly enforced at both the representation level and the gradient level. Finally, the IDFL yields disentangled seasonal–trend features that provide accurate prediction.

**Decomposition Module.** To begin with, the time series is decomposed into seasonal and trend features. Given a time series with $C$ features $\mathcal{T} \in \mathbb{R}^{B \times L \times C}$, we extract the trend component by a moving-average kernel of length $k_{trend}$:

$$\mathbf{t} = AvgPool_{k_{trend}}(\mathcal{T}), \mathbf{s} = \mathcal{T} - \mathbf{t}, \tag{1}$$

where $\mathbf{t} \in \mathbb{R}^{B \times L \times C}$ and $\mathbf{s} \in \mathbb{R}^{B \times L \times C}$ denote the trend and seasonal signals, respectively.

To diversify the decomposed trend and seasonal components, we perform stochastic forward passes by applying dropout in the decomposition module and input the same time series $\mathcal{T}$ into the decomposition module twice:

$$\mathbf{s}^{(1)}, \mathbf{t}^{(1)} = Decomposition(\mathcal{T}), \mathbf{s}^{(2)}, \mathbf{t}^{(2)} = Decomposition(\mathcal{T}). \tag{2}$$

Next, features $(\mathbf{s}^{(1)}, \mathbf{s}^{(2)})$ and $(\mathbf{t}^{(1)}, \mathbf{t}^{(2)})$ are separately fed into the forecasters.

**Forecaster.** The forecaster is implemented as a lightweight Time-Series Feature Extractor (TSFE) (Miao et al., 2024), consisting of patching with patch length $P$ and stride $S$, $N_{op}$ stacked self-attention and feed-forward networks, and a linear layer that outputs latent features:

$$\mathbf{z}_{tre}^{(i)} = Forecaster_{tre}(\mathbf{t}^{(i)}), \ \mathbf{z}_{sea}^{(i)} = Forecaster_{sea}(\mathbf{s}^{(i)}), \tag{3}$$

where $i \in \{1, 2\}$ and $\mathbf{z}_{tre}, \mathbf{z}_{sea} \in \mathbb{R}^{B \times N \times C}$ is the prediction of trend and seasonal features.

Given the mean squared error loss function $l(\cdot, \cdot)$, the time seires forecasting loss can be defined as:

$$\mathcal{L} = l(\mathbf{z}_{tre} + \mathbf{z}_{sea}, y), \tag{4}$$

where $y$ represents the ground truth for the input time series.

**Representation-Level Alignment.** Representation-level alignment is designed to force samples that share a pattern, no matter which domain they come from, to occupy the same region of the feature space. Concretely, the network learns a single mapping that pushes every domain's distribution toward one common statistical form. Here, we denote the forecasting task set as $\mathcal{C} = \{seasonal, trend\}$. We use $\mathbf{s}, \bar{\mathbf{s}}$ to represent different seasons, and use $\mathbf{t}, \bar{\mathbf{t}}$ to represent different trends so that we can denote decomposed features as $\mathcal{F} = \{(\mathbf{s}, \mathbf{t}), (\mathbf{s}, \bar{\mathbf{t}}), (\mathbf{t}, \mathbf{s}), (\mathbf{t}, \bar{\mathbf{s}})\}$, where $\{(\mathbf{s}, \mathbf{t}), (\mathbf{s}, \bar{\mathbf{t}})\}$ represent $\mathbf{s}^{(1)}, \mathbf{s}^{(2)}, \{(\mathbf{t}, \mathbf{s}), (\mathbf{t}, \bar{\mathbf{s}})\}$ represent $\mathbf{t}^{(1)}, \mathbf{t}^{(2)}$. For example, $(\mathbf{s}, \mathbf{t})$ represents a seasonal feature with a different trend than another seasonal feature $(\mathbf{s}, \bar{\mathbf{t}})$, considering that seasonal and trend features have not yet been fully disentangled.

Initially, we derive the gradient of the model on each branch with respect to the representations:

$$g_j^i = \frac{\partial(Forecaster_j(Decomposition(\mathcal{T}))}{\mathcal{X}}, \tag{5}$$

where $i \in \mathcal{F}, j \in \mathcal{C}$. $\mathcal{X}$ denotes the embedding of input time series $\mathcal{T}$.

The representations associated with similar gradients indicate intrinsic characteristics of seasonal patterns that are invariant to trend factors or vice versa. Consequently, we compute the absolute value of the difference between the two gradients:

$$\Delta g_{sea} = |g_{sea}^{(\mathbf{s,t})} - g_{sea}^{(\mathbf{s,\bar{t}})}|, \Delta g_{tre} = |g_{tre}^{(\mathbf{t,s})} - g_{tre}^{(\mathbf{t,\bar{s}})}|. \tag{6}$$

The variables with a small difference correspond to seasonal features that are insensitive to trend variation and trend features that are insensitive to seasonal variation. We rank the absolute gradient differences in descending order and then take the $\alpha$-percentile value, denoted as $d^\alpha$. A binary mask $m$ of identical shape to the representation is then generated. For the $k$-th element,

$$m_j(k) = \begin{cases} 0, & \Delta g_j(k) \geq d^\alpha \\ 1, & else \end{cases}. \tag{7}$$

By applying the mask to the original representation, the network filters out component-varying feature variables to learn the invariant seasonal feature $\hat{\mathbf{s}}$ and invariant trend feature $\hat{\mathbf{t}}$,

$$\hat{\mathbf{s}} = \mathcal{X} \odot m_{sea}, \hat{\mathbf{t}} = \mathcal{X} \odot m_{tre}. \tag{8}$$

Then, the learned invariant features are fed into the forecaster:

$$\hat{\mathbf{z}}_{tre} = Forecaster_{tre}(\hat{\mathbf{t}}), \hat{\mathbf{z}}_{sea} = Forecaster_{sea}(\hat{\mathbf{s}}). \tag{9}$$

Finally, the mean square error is defined as the loss of invariant features in predictions,

$$\mathcal{L}_{inv} = l(\hat{\mathbf{z}}_{tre} + \hat{\mathbf{z}}_{sea}, y), \tag{10}$$

where $y$ represents the ground truth of the input time series.

**Fourier Transform Module.** Average pooling provides a coarse time-domain decomposition into trend and residual, while fourier transform offers a fine-grained frequency-domain analysis to capture periodic components. This module aims to provide frequency-consistent supervision for invariant learning. Time-series windows are decomposed in the time domain via the Discrete Fourier Transform (DFT). Given the whole time series embedding $\mathcal{X} \in \mathbb{R}^{B \times L \times E}$, we treat each channel independently. The DFT of each channel is defined as:

$$\mathbf{X}[k] = \sum_{t=0}^{L-1} \mathbf{X}[t] \exp\left(-\frac{2\pi i}{L}kt\right), \quad k = 0, \dots, L-1. \tag{11}$$

Specifically, frequency coefficients are split into low-frequency (trend) and high-frequency (seasonality) subsets using a predefined cut-off index $k_{cut}$:

$$\mathbf{X}_{\mathrm{tr}}[k] = \begin{cases} \mathbf{X}[k], & 0 \leq k \leq k_{cut}, \\ 0, & otherwise, \end{cases} \quad \mathbf{X}_{sea}[k] = \begin{cases} \mathbf{X}[k], & k_{cut} < k \leq \lfloor L/2 \rfloor, \\ 0, & otherwise. \end{cases} \tag{12}$$

Then, the inverse DFT $\mathcal{F}^{-1}(\cdot)$ is applied to obtain the trend $\mathbf{t}$ and seasonal signals $\mathbf{s}$, respectively:

$$\mathbf{t} = \mathcal{F}^{-1}(\mathbf{X}_{tr}), \quad \mathbf{s} = \mathcal{F}^{-1}(\mathbf{X}_{sea}). \tag{13}$$

After the representation-level alignment, we fed the predictions of invariant features into Fourier transform module $FT(\cdot)$, getting the decomposed seasonal and trend features:

$$\mathbf{s}', \mathbf{t}' = FT(\hat{\mathbf{z}}_{tre} + \hat{\mathbf{z}}_{sea}). \tag{14}$$

Finally, we compute the loss function with the new decomposed features and the prediction of raw decomposed features:

$$\mathcal{L}_{pred} = \sum_{i=1}^{2} l(Forecaster_{sea}(\mathbf{s}^{(i)}), \mathbf{s}') + \sum_{i=1}^{2} l(Forecaster_{tre}(\mathbf{t}^{(i)}), \mathbf{t}'). \tag{15}$$

The loss function $\mathcal{L}_{rep}$ is the combination of the trend-irrelevant seasonal-specific representation and the seasonal-irrelevant trend-specific representation:

$$\mathcal{L}_{rep} = l(Forecaster_{tre}(\hat{\mathbf{t}}), \mathbf{t}') + l(Forecaster_{sea}(\hat{\mathbf{s}}), \mathbf{s}'). \tag{16}$$

**Gradient-Level Alignment.** Gradient-level alignment aims to optimize the trajectories of all branches toward a common direction. By explicitly shrinking the dispersion of inter-branch gradients, the model is encouraged to discard component-specific cues and retain invariant ones. We derive the gradient of seasonal predictions with respect to the seasonal forecaster under varying trends, that of trend predictions under varying seasonals, as detailed below:

$$G_{sea}^i = \frac{\partial l(Forecaster_{sea}(\mathbf{s}^i), \mathbf{s}')}{\partial \theta_{sea}}, G_{tre}^i = \frac{\partial l(Forecaster_{tre}(\mathbf{t}^i), \mathbf{t}')}{\partial \theta_{tre}}, \tag{17}$$

where $\theta$ denotes the parameters of $Forecaster(\cdot)$ and decomposition module $Dcomposition(\cdot)$.

Steering every branch along this identical route markedly eases the acquisition of invariant predictions (Chen et al., 2025b). To enforce this gradient-level alignment and distill disentangled invariances, we suppress the model's ability to identify patterns by minimizing the Euclidean distance, denoted $d_{euc}(\cdot, \cdot)$, between the respective gradient vectors, formulated as:

$$\mathcal{L}_{grad} = d_{euc}(G_{sea}^{(s,t)}, G_{sea}^{(s,\bar{t})}) + d_{euc}(G_{tre}^{(t,s)}, G_{tre}^{(t,\bar{s})}). \tag{18}$$

Therefore, the gradient-level alignment drives all parameter updates along a unified trajectory, thereby strengthening the robustness of the forecaster.

## 3.2 PROXY DENOISING

LLMs for time series modeling benefit from their pre-trained knowledge and generation ability, but would introduce hallucinations. The proxy denoising (PD) is proposed to quantify the proxy error of the LLMs and then generate calibrated predictions for enhancing prediction. It leverages the disagreement between the source model $\theta_s$ and the target model $\theta_t$ to estimate and suppress the noise dynamically. $\theta_s$ encodes knowledge acquired on the source distribution, remaining oblivious to target-specific drift, while $\theta_t$ is trainable and gradually adapts to the target. Its current state reflects the best in-domain hypothesis available at any moment.

When all three models agree, the LLM is likely reliable. If $\theta_s$ and $\theta_t$ agree with each other but deviate from the LLM, the discrepancy is interpreted as proxy noise that needs to be corrected. For every target mini-batch $B_t = \left\{x_i\right\}_{i=1}^{B}$, we compute the prediction of three models: $z_{ts,i} = \theta_{ts}(x_i), z_{s,i} = \theta_s(x_i), z_{t,i} = \theta_t(x_i)$. The per-sample noise vector is simply the signed residual $e_i = \theta_s(x_i) - \theta_t(x_i)$, which captures how far the LLM predictions deviate from the consensus of source and target models. The subtraction serves as an empirical error signal. The estimated noise is subtracted from the LLM outputs to obtain the denoised prediction:

$$\tilde{z}_i = \theta_{ts}(x_i) - \alpha(\theta_s(x_i) - \theta_t(x_i)), \tag{19}$$

where $\theta_s, \theta_t, \theta_{ts}$ apply the source model, target model, and LLM to get the corresponding prediction, and $\alpha$ represents the correction strength, which is a hyperparameter. Particularly, $\alpha = 1$ performs full correction (complete trust in the source-target consensus) and $\alpha = 0$ retains the raw LLM predictions. The denoised predictions $\tilde{z}_i$ are forwarded to the subsequent knowledge distillation.

## 3.3 KNOWLEDGE DISTILLATION

To improve inference efficiency, LLM's outputs are distilled to a lightweight target model to guide the model with purified knowledge and prevent the LLM from drifting away from the target domain. The output of the target model is aligned with the corrected proxy via Mean Squared Error:

$$\mathcal{L}_{kd} = l(\theta_{ts}(x_i) - \alpha(\theta_s(x_i) - \theta_t(x_i)), \theta_t(x_i)). \tag{20}$$

Minimizing $\mathcal{L}_{kd}$ pulls the target model's predictions toward the denoised large language model without any label supervision. The gradient flow is one-way: only the target model $\theta_t$ is updated; the large language model remains frozen. Consequently, the target model receives high-level temporal knowledge distilled from the denoised LLM while preserving its own low-rank adaptation capacity.

### 3.4 OVERALL OBJECTIVE FUNCTION

The final loss consists of a time series forecasting loss $\mathcal{L}$, an invariant features forecasting loss $\mathcal{L}_{inv}$, a disentangled features forecasting loss $\mathcal{L}_{pred}$, a representation invariant loss $\mathcal{L}_{rep}$, a gradient invariant loss $\mathcal{L}_{grad}$ and a knowledge distillation loss $\mathcal{L}_{kd}$. We combine them together, and the overall loss is:

$$\begin{aligned} \mathcal{L}_{all} = \mathcal{L} &+ \lambda_{inv}\mathcal{L}_{inv} + \lambda_{pred}\mathcal{L}_{pred} \\ &+ \lambda_{rep}\mathcal{L}_{rep} + \lambda_{grad}\mathcal{L}_{grad} + \lambda_{kd}\mathcal{L}_{kd}, \end{aligned} \tag{21}$$

where $\lambda_{inv}, \lambda_{pred}, \lambda_{rep}, \lambda_{grad}, \lambda_{kd}$ are trade-off parameters.

## 4 EXPERIMENTS

### 4.1 EXPERIMENTAL SETUP

The experiments are carried out on six widely-used time series datasets, including ETTh1, ETTh2, ETTm1, ETTm2, Weather, Traffic, and Electricity (Liu et al., 2024b). We compare TimePD with the following existing baselines: DLinear (Zeng et al., 2023), TimeKAN (Huang et al., 2025), SimpleTM (Chen et al., 2025a), TimesNet (Wu et al., 2023), TimeMixer (Wang et al., 2024b), WPMixer (Murad et al., 2025), iTransformer (Liu et al., 2024c), FEDformer (Zhou et al., 2022), PatchTST (Nie et al., 2023), OFA (Zhou et al., 2023) and TimeLLM (Jin et al., 2024). We provide more details of datasets and baselines in Appendix A.2. For the source-free training, we first train a source model on a source dataset (e.g., ETTh1). Then, we initialize the target model with the trained source model and then train the target model on the target domain (e.g., Weather). For example, we use ETTh1 $\rightarrow$ Weather to denote a source-free training process where the source and target domains are ETTh1 and Weather, respectively. Time series foundation models are not compared due to different objectives, where foundation models aim to achieve broad generalization by pretraining on massive multi-domain data. To enable fair comparison, we train the baselines on the source data and then finetune them with 30% of the target data for testing. We employ the stacked TSFE as the forecaster (Miao et al., 2024), and OFA as the LLM backbone. Mean Absolute Error (MAE) and Mean Square Error (MSE) are adopted as the evaluation metrics.

We implement our model using PyTorch on the NVIDIA A800 GPU. The hyperparameters in the model are set as follows. Target-domain dataset size is 30%. The weight of rep losses at the representation level and grad losses at the gradient level are set to 1/8 and 1/2, respectively. The weight of loss at knowledge distillation is set to 0.001. The dropout rate in the decomposition block is set to 0.1. The patch length and stride are set to 16 and 8, respectively. The initial learning rate is 0.0001. ETT datasets and other datasets are split into the training data, validation data, and test data by the ratios of 6:2:2 and 7:1:2, respectively. The parameters of the baseline methods are set according to their original papers and any accompanying code. All of the models follow the same experimental setup with prediction length $PL \in \{96, 192, 336\}$ on all datasets.

### 4.2 EXPERIMENTAL RESULTS

#### 4.2.1 OVERALL PERFORMANCE COMPARISON

We evaluate the source-free long-term forecasting capabilities of TimePD and baselines on six datasets (ETTh2, ETTm1, ETTm2, Weather, Electricity, Traffic) transferred from ETTh1 in Table 2. Results on other source datasets are provided in A.3.1. The best performance is marked in bold, and the second-best performance is underlined. From the comparison results, it is evident that TimePD achieves the best performance on most datasets across all prediction lengths. Averaged across all 18 tasks (6 datasets $\times$ 3 prediction lengths), TimePD obtains the lowest average MSE and MAE, outperforming the most advanced method (Time-LLM and OFA) with an average MSE reduction by 4.98% and 4.39%, and MAE reduction by 2.64% and 3.21%, respectively. The most substantial improvements are observed on the ETTh1 $\rightarrow$ Weather and ETTh1 $\rightarrow$ ETTh2 datasets, particularly at shorter horizons (e.g., PL = 96), where TimePD reduces MSE by over 10.00% and MAE by over 5.03% compared to Time-LLM. This is due to that TimePD learns the invariant features contained in time series through IDFL, and leverages LLM, denoised via proxy denoising, to guide the target model. Moreover, TimePD is particularly effective on complex datasets. On ETTh1 $\rightarrow$ traffic

Table 1: Overall Performance Comparison.

| Methods | Dataset | ETTh1 → ETTh2 | | | ETTh1 → ETTm1 | | | ETTh1 → ETTm2 | | | ETTh1 → Weather | | | ETTh1 → Electricity | | | ETTh1 → Traffic | | |
|---|---|---|---|---|---|---|---|---|---|---|---|---|---|---|---|---|---|---|---|
| | PL | 96 | 192 | 336 | 96 | 192 | 336 | 96 | 192 | 336 | 96 | 192 | 336 | 96 | 192 | 336 | 96 | 192 | 336 |
| DLinear | MSE | 0.287 | 0.367 | 0.438 | **0.357** | 0.396 | 0.428 | 0.180 | 0.240 | 0.301 | 0.178 | 0.220 | **0.262** | 0.178 | 0.191 | 0.217 | 0.460 | 0.484 | 0.522 |
| | MAE | 0.346 | 0.400 | 0.453 | 0.391 | 0.422 | 0.443 | 0.274 | 0.319 | 0.363 | 0.244 | 0.280 | 0.314 | 0.279 | 0.292 | 0.319 | 0.332 | 0.348 | 0.377 |
| TimeKAN | MSE | 0.284 | 0.352 | 0.409 | 0.367 | 0.399 | 0.426 | 0.182 | 0.236 | 0.280 | 0.735 | 0.742 | 0.745 | 1.085 | 1.083 | 1.079 | 0.540 | 0.555 | 0.557 |
| | MAE | 0.343 | 0.387 | 0.429 | 0.399 | 0.417 | 0.431 | 0.267 | 0.303 | 0.329 | 0.665 | 0.666 | 0.666 | 0.853 | 0.852 | 0.852 | 0.828 | 0.831 | 0.829 |
| SimpleTM | MSE | 0.283 | 0.351 | 0.355 | 0.383 | 0.401 | 0.486 | 0.182 | 0.231 | 0.278 | 0.754 | 0.768 | 0.701 | 1.080 | 1.074 | 1.074 | 0.526 | 0.539 | 0.549 |
| | MAE | 0.340 | 0.388 | 0.404 | 0.399 | 0.414 | 0.464 | 0.269 | 0.302 | 0.333 | 0.670 | 0.679 | 0.645 | 0.851 | 0.851 | 0.851 | 0.825 | 0.828 | 0.829 |
| TimesNet | MSE | 0.362 | 0.429 | 0.457 | 0.498 | 0.635 | 0.617 | 0.213 | 0.270 | 0.315 | 0.737 | 0.742 | 0.744 | 1.086 | 1.082 | 1.080 | 0.524 | 0.534 | 0.546 |
| | MAE | 0.408 | 0.437 | 0.461 | 0.462 | 0.534 | 0.526 | 0.297 | 0.331 | 0.358 | 0.665 | 0.665 | 0.668 | 0.853 | 0.852 | 0.852 | 0.825 | 0.827 | 0.828 |
| TimeMixer | MSE | 0.330 | 0.399 | 0.431 | 0.438 | 0.534 | 0.491 | 0.185 | 0.234 | 0.282 | 0.744 | 0.741 | 0.743 | 1.083 | 1.083 | 1.082 | 0.524 | 0.534 | 0.548 |
| | MAE | 0.373 | 0.413 | 0.447 | 0.425 | 0.483 | 0.461 | 0.269 | 0.303 | 0.333 | 0.667 | 0.665 | 0.666 | 0.852 | 0.852 | 0.852 | 0.825 | 0.827 | 0.829 |
| WPMixer | MSE | 0.291 | 0.375 | 0.414 | 0.371 | 0.403 | 0.461 | 0.182 | 0.233 | 0.284 | 0.736 | 0.764 | 0.741 | 1.082 | 1.083 | 1.080 | 0.525 | 0.536 | 0.548 |
| | MAE | 0.348 | 0.399 | 0.432 | 0.392 | **0.409** | 0.441 | 0.270 | 0.302 | 0.337 | 0.663 | 0.677 | 0.665 | 0.852 | 0.852 | 0.852 | 0.825 | 0.827 | 0.829 |
| iTransformer | MSE | 0.358 | 0.489 | 0.510 | 0.402 | 0.420 | 0.561 | 0.194 | 0.237 | 0.312 | 0.189 | 0.237 | 0.274 | 0.180 | 0.191 | 0.219 | 0.452 | 0.479 | 0.516 |
| | MAE | 0.395 | 0.467 | 0.487 | 0.415 | 0.426 | 0.503 | 0.275 | 0.306 | 0.351 | 0.238 | 0.282 | 0.309 | 0.288 | 0.299 | 0.316 | 0.341 | 0.358 | 0.387 |
| FEDformer | MSE | 0.388 | 0.485 | 0.421 | 0.646 | 0.630 | 0.695 | 0.287 | 0.331 | 0.391 | 0.747 | 0.746 | 0.734 | 1.083 | 1.082 | 1.079 | 0.529 | 0.535 | 0.548 |
| | MAE | 0.431 | 0.500 | 0.462 | 0.534 | 0.544 | 0.566 | 0.359 | 0.388 | 0.423 | 0.667 | 0.668 | 0.661 | 0.852 | 0.852 | 0.851 | 0.827 | 0.827 | 0.829 |
| PatchTST | MSE | 0.280 | 0.359 | 0.354 | 0.364 | 0.400 | 0.430 | 0.180 | 0.234 | 0.286 | 0.232 | 0.279 | 0.335 | 0.181 | 0.197 | 0.216 | 0.455 | 0.484 | 0.519 |
| | MAE | 0.340 | 0.388 | 0.399 | 0.398 | 0.420 | 0.438 | 0.267 | 0.303 | 0.338 | 0.282 | 0.318 | 0.357 | 0.278 | 0.292 | 0.313 | 0.333 | 0.345 | 0.382 |
| OFA | MSE | 0.296 | 0.374 | 0.394 | 0.382 | 0.404 | 0.430 | 0.185 | 0.231 | 0.287 | 0.195 | 0.232 | 0.269 | 0.173 | 0.206 | 0.220 | 0.459 | 0.483 | 0.515 |
| | MAE | 0.352 | 0.404 | 0.424 | 0.399 | 0.413 | 0.433 | 0.270 | 0.305 | 0.338 | 0.248 | 0.280 | 0.304 | 0.283 | 0.313 | 0.320 | 0.347 | 0.356 | 0.380 |
| Time-LLM | MSE | 0.324 | 0.374 | 0.393 | 0.402 | 0.424 | 0.456 | 0.183 | 0.236 | 0.283 | 0.176 | 0.224 | 0.272 | 0.171 | 0.192 | 0.220 | 0.452 | 0.478 | 0.513 |
| | MAE | 0.367 | 0.400 | 0.421 | 0.410 | 0.423 | 0.434 | 0.272 | 0.307 | 0.331 | 0.229 | 0.275 | 0.304 | 0.284 | 0.291 | 0.318 | 0.338 | 0.347 | 0.381 |
| TimePD | MSE | **0.280** | **0.345** | **0.346** | 0.359 | **0.392** | **0.422** | **0.177** | **0.230** | **0.277** | **0.169** | **0.219** | **0.265** | **0.170** | **0.187** | **0.211** | **0.452** | **0.474** | **0.510** |
| | MAE | **0.338** | **0.385** | **0.398** | **0.390** | 0.413 | **0.430** | **0.267** | **0.297** | 0.330 | **0.228** | **0.275** | **0.303** | **0.276** | **0.289** | **0.312** | **0.327** | **0.342** | **0.373** |

dataset, where baseline methods suffer from high variance due to noise and periodicity. For example, at PL = 96, TimePD reduces MAE by 3.25% compared to Time-LLM. The results demonstrate that TimePD has a superior generalization ability.

### 4.2.2 ABLATION STUDY

To assess the contribution of each component in TimePD, we evaluate three invariants: (1) *w/o_LLM*: TimePD without the large language model; (2) *w/o_PD*: TimePD without the proxing denoising; and (3) *w/o_KD*. TimePD without the knowledge distillation and report ablation study results in Figure 4 and A.3.2. The most substantial performance drop is observed when *w/o_KD* is removed. For example, on ETTh1 → ETTh2, the MSE rises by 17.86% and MAE rises by 18.34%. The reason is that knowledge distillation proves to be the most critical, as it effectively transfers domain-shared temporal knowledge from the LLM to the target model. The removal of *w/o_PD* leads to further degradation since it refines the LLM outputs by mitigating noise. On ETTh1 → ETTm1, MSE increases sharply by 26.18%, and MAE increases by 12.31%. We also observe that performance degrades moderately across all datasets under *w/o_LLM*, as it serves as an indispensable knowledge source. For instance, on ETTh1 → ETTh2, MSE increases by 4.29% and MAE increases by 3.55%.

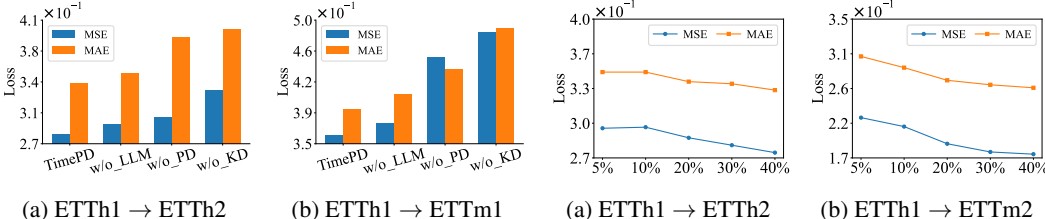

(a) ETTh1 → ETTh2    (b) ETTh1 → ETTm1

(a) ETTh1 → ETTh2    (b) ETTh1 → ETTm2

Figure 4: Performance of TimePD and its variants.

Figure 5: Effect of target dataset size.

### 4.2.3 EFFECT OF THE SIZE OF THE TARGET DOMAIN DATASET

To verify the scalability of TimePD, we conduct experiments using 5%, 10%, 20%, 30%, and 40% of the dataset. We observe that increasing the proportion of the target domain dataset generally improves performance in terms of MSE and MAE. Specifically, on the ETTh1 → ETTh2 and ETTh1 → ETTm2 tasks, the model exhibits a consistent decrease in both MSE and MAE as more target domain data is used, as shown in Figure 5. This suggests that these tasks benefit significantly from domain-specific supervision and that the TimePD is capable of effectively leveraging more target samples. More scalability analysis results are presented in Appendix A.3.3.

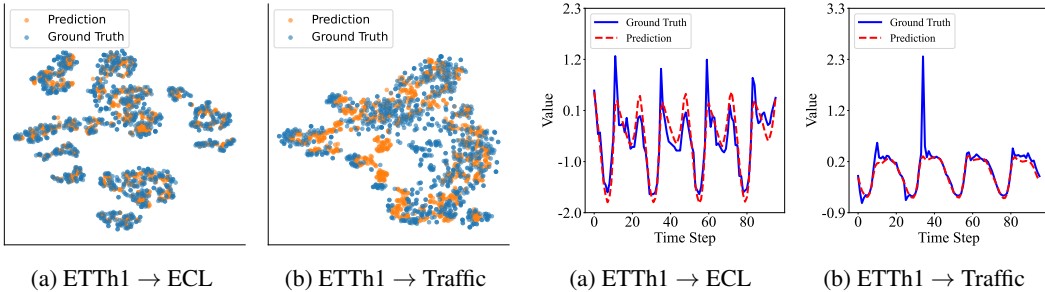

| (a) ETTh1 → ECL | (b) ETTh1 → Traffic | (a) ETTh1 → ECL | (b) ETTh1 → Traffic |

Figure 6: Data distribution visualization.  Figure 7: Prediction vs. Ground Truth.

### 4.2.4 DATA DISTRIBUTION VISUALIZATION OF PREDICTION AND GROUND TRUTH

To validate whether the prediction follows the similar distribution as the ground truth, we visualize the t-SNE (Maaten & Hinton, 2008) across ETTh1, Traffic, and ECL datasets, as shown in Figure 6. 1000 prediction–ground truth pairs are randomly sampled. Across all subfigures, the predicted values (orange) closely align with the true values (blue), indicating that TimePD effectively captures the underlying data distribution. Furthermore, we observe that the spatial structure of the data clusters is also well preserved between predictions and ground truth.

### 4.2.5 CASE STUDY ON PREDICTION CONSISTENCY

To further intuitively demonstrate the effectiveness of the proposed TimePD, we compare the predictions with ground truth on ETTh1 → ECL and ETTh1 → Traffic settings. As shown in Figure 7, the predicted curves closely follow the actual trajectories, capturing both periodic trends and abrupt variations. On ECL, the model reproduces seasonal peaks and troughs with minor deviations at sharp transitions. On the Traffic dataset, the predictions remain well aligned with sudden spikes, demonstrating robustness across domains. These results qualitatively confirm that TimePD generalizes well and produces reliable forecasts beyond quantitative metrics.

### 4.2.6 INVARIANT FEATURE VISUALIZATION

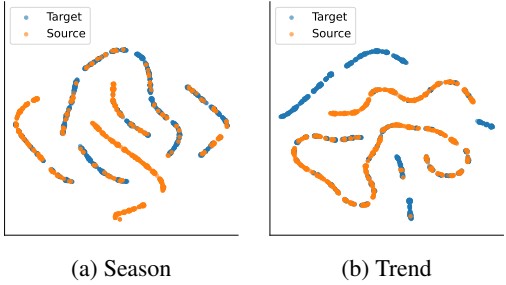

| (a) Season | (b) Trend |

Figure 8: Invariant feature visualization.

To test whether TimePD is effective in capturing the invariant features, especially the seasonal and trend information, we visualize such frequency information of source data and target data in one figure with t-SNE. The results on two datasets are shown in Figure 8, where Figures 8 (a) and (b) show the season and trend comparison on the ETT dataset. Blue and orange dots represent the target data and source data, respectively. We observe that blue dots almost follow the trace of orange dots, indicating that the models learns the invariant features, i.e., season and trend information. Although some exceptions exist in the trend comparison, these show that TimePD not only learns the invariant features between source and target data but also is capable of extracting specialized features that are useful for the target time series forecasting.

## 5 CONCLUSION

We present TimePD, a new source-free time series forecasting framework with proxy denoising that unleashes the power of LLMs and sufficient knowledge extracted from the source domain without accessing its raw data. To enable effective temporal correlation capturing and alleviate concept drift across domains, we propose an invariant disentangled feature learning module based on a dual-branch architecture. Further, a proxy denoising mechanism is proposed to dynamically incorporate the generalized knowledge learned by LLMs, enhancing model performance. We also employ the knowledge distillation to calibrate the final prediction with denoised prediction. An empirical study on real datasets offers evidence that the paper's proposals improve on the state-of-the-art in terms of prediction accuracy. An interesting research direction is to attempt to apply the proposed TimePD to other time series related tasks, e.g., anomaly detection.

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

# A APPENDIX

## A.1 PRELIMINARY

This section introduces the basic concepts, notations, and preliminaries that underpin the proposed TimePD framework. We first formalize the source-free time-series forecasting problem under domain shift, and then establish a theory of invariant disentangled features for domain robustness, and introduce a proxy confidence theory for bias correction in large language models. Throughout, we adopt the following general notation: Bold lower-case symbols (e.g., $\mathbf{x}, \mathbf{y}$) denote vectors; bold upper-case symbols (e.g., $\mathbf{X}, \mathbf{Y}$) denote matrices or higher-order tensors. Calligraphic symbols (e.g., $\mathcal{X}, \mathcal{Y}$) denote sets or distributions. Subscripts $s$ and $t$ indicate source and target domains, respectively. Unless stated otherwise, all norms $\|\cdot\|$ are L2 norms.

### A.1.1 SOURCE-FREE TIME-SERIES FORECASTING

Let $\mathcal{T}$ denote a multivariate time-series sample of length $L$ and channel (feature) dimension $C$: $\mathcal{T} \in \mathbb{R}^{L \times C}$. Given a look-back window of length $l$: $\mathbf{x} = \mathcal{T}\{t - l + 1 : t, \cdot\} \in \mathbb{R}^{l \times C}$ the forecasting task is to predict the next $H$ steps: $\mathbf{y} = \mathcal{T}\{t + 1 : t + H, \cdot\} \in \mathbb{R}^{H \times C}$.

**Domains.** Source domain $\mathcal{D}_s = \{\mathcal{X}_s, \mathcal{Y}_s\}$ contains labeled pairs $(\mathbf{x}_s, \mathbf{y}_s)$ drawn from distribution $\mathcal{P}_s(\mathcal{X}, \mathcal{Y})$. Target domain $\mathcal{D}_t = \{\mathcal{X}_t\}$ contains unlabeled samples $\mathbf{x}_t$ from distribution $\mathcal{P}_t\{\mathcal{X}\}$, where $\mathcal{P}_t \neq \mathcal{P}_s$ .

**Source-free constraint.** At adaptation time, the original source data $\{\mathbf{x}_s, \mathbf{y}_s\}$ are inaccessible due to privacy or legal constraints. Only a pre-trained source model $\theta_s$ (parameterized by $\phi_s$) is available. The goal is to learn a target model $\theta_t$ (parameterized by $\phi_t$) that minimizes the expected forecast error on $\mathcal{D}_t$:

$$min_{\phi_t} \mathbb{E}_{\mathbf{x} \sim \mathcal{P}_t}[\mathcal{L}(\theta_t(\mathbf{x}), \mathbf{y})], \tag{22}$$

where $\mathbf{y}$ is the (unknown) ground-truth future values, and $\mathcal{L}(\cdot, \cdot)$ is a loss function (e.g., MSE).

### A.1.2 INVARIANT DISENTANGLED FEATURES

Domain shifts in time series manifest as perturbations to trend $\mathcal{T}_{tre}$ or seasonality $\mathcal{S}_{sea}$. An invariant feature $\phi^*$ satisfies:

$$\phi^*(x) \approx \phi^*(x') \quad \forall x, x' \text{ s.t. } x \in D_i, \ x' \in D_j, \ C(x) = C(x'), \tag{23}$$

where $\mathcal{D}_i, \mathcal{D}_j$ are domains (e.g., differing trend contexts), and $\mathcal{C}$ denotes the component class (e.g., seasonal pattern). Disentanglement requires that features are component-specific: $\phi_s(x)$ encodes seasonality and is invariant to trend variations $\Delta t$, and $\phi_t(x)$ encodes trend and is invariant to seasonal variations $\Delta s$. Formally, for small perturbations $\epsilon$:

$$\begin{aligned}
\left\|\phi_s(s + t + \varepsilon\Delta t) - \phi_s(s + t)\right\|_2 &< \delta_s, \\
\left\|\phi_t(s + t + \varepsilon\Delta s) - \phi_t(s + t)\right\|_2 &< \delta_t,
\end{aligned} \tag{24}$$

where $\delta_s, \delta_t \to 0$ for perfect invariance. Learning such features necessitates suppressing gradient pathways sensitive to cross-component variations, enabling generalization across domains where either component shifts (Parascandolo et al., 2021).

### A.1.3 PROXY CONFIDENCE THEORY

When a pre-trained large language model is used as a proxy forecaster on the unlabeled target domain, its outputs inevitably deviate from the latent "domain-invariant" distribution because of domain shift. We therefore treat the large language model as a noisy proxy and quantify its reliability through a proxy confidence theory.

**Notations.** $\mathcal{D}_S$ represents the source domain distribution (known only via the pretrained source model $\theta_s$). $\mathcal{D}_T^t$ represents target model $\theta_t$ distribution at adaptation step t. $\mathcal{D}_{TS}$ represents proxy (LLM $\theta_{ts}$) distribution. $\mathcal{D}_l$ represents latent domain-invariant distribution.

**Proxy Error.** Define the proxy error at step t as the expected divergence between the proxy and the ideal space:

$$e_t = \mathbb{E}_{x \sim \mathcal{D}_T}[D(\theta_{ts}(x), \theta_l(x))], \tag{25}$$

where $D(\cdot, \cdot)$ is a distance in logit space. Since $\theta_l$ is inaccessible, we approximate $e_t$ by the disagreement between source and target models:

$$e_t \approx \mathbb{E}_{x \sim \mathcal{D}_T}[\|\theta_s(x) - \theta_t(x)\|_2]. \tag{26}$$

Larger disagreement $\Rightarrow$ larger proxy error.

**Proxy Confidence Score.** We map the error to a confidence weight:

$$\mathcal{C}_t = exp(-e_t/\tau) \in (0, 1], \tag{27}$$

with temperature $\tau > 0$. At $t = 0, \theta_t \approx \theta_s \Rightarrow e_t \approx 0 \Rightarrow \mathcal{C}_t \approx 1$(high trust), As adaptation proceeds, $\theta_t$ drifts from $\theta_s \Rightarrow e_t$ grows $\mathcal{C}_t \downarrow$(reduced trust).

The proxy confidence theory thus provides an online, parameter-free mechanism to quantify and mitigate the noise inherent in large language model forecasts during source-free domain adaptation.

## A.2 EXPERIMENTAL SETUP DETAILS

### A.2.1 DATASETS.

We conduct comprehensive experiments under a source-free domain adaptation scenario using seven widely-used time series datasets, covering four application domains: weather, traffic, economics, and energy.

- **Weather.** The Weather dataset contains 21 indicators of weather(e.g., air temperature and humidity), which are collected in Germany. The data is recorded every 10 minutes.
- **Traffic.** The Traffic dataset contains hourly road occupancy rates obtained from sensors located on San Francisco freeways from 2015 to 2016.

- **Electricity.** The Electricity dataset contains the hourly electricity consumption of 321 clients from 2012 to 2014.

- **ETT.** The ETT dataset includes two hourly-level datasets(ETTh1 and ETTh2) and two 15-minute-level datasets (ETTm1 and ETTm2). Each dataset includes 7 oil and load features of electricity transformers between July 2016 and July 2018.

We chose time series forecasting as a representative downstream task, as it is a popular analytics task. We employ the proposed stacked TSOperators as the forecasting models in the source model and target model, and employ OFA as the LLM backbone.

### A.2.2 BASELINES.

We compare TimePD with the following existing methods for time series forecasting.

- **OFA:** Introduces a frozen pretrained Transformer framework that reuses frozen self-attention and feedforward blocks from large pretrained language or vision models, fine-tuning only lightweight adapters to achieve state-of-the-art performance across diverse time series tasks such as forecasting, classification, and anomaly detection (Zhou et al., 2023).

- **SimpleTM:** Introduces a simple yet effective architecture that uniquely integrates classical signal processing ideas with a slightly modified attention mechanism (Chen et al., 2025a).

- **TimeKAN:** Employs a kernel attention network to decompose the time series into frequency components and model each for improved long-term forecasting (Huang et al., 2025).

- **TimeMixer:** Utilizes a decomposable multiscale mixing module to integrate information across different temporal scales for more robust predictions (Wang et al., 2024b).

- **iTransformer:** Introduces an "inverted" transformer architecture that swaps the roles of queries, keys, and values to simplify and accelerate time series modeling (Liu et al., 2024c).

- **PatchTST:** Treats a time series as a sequence of fixed-length patches and applies transformer-based patch-wise modeling to capture long-range dependencies (Nie et al., 2023).

- **TimesNet:** Constructs a 2D temporal-variation representation and applies joint time–frequency convolutions to capture general patterns in time series data (Wu et al., 2023).

- **DLinear:** Decomposes the series into trend and seasonal components, fits each with a simple linear model, and then recombines them for forecasting (Zeng et al., 2023).

- **FEDformer:** Leverages frequency-enhanced decomposition within a transformer framework to efficiently model and forecast long-term periodic patterns (Zhou et al., 2022).

- **WPMixer:** This method is an MLP-based model that performs multi-resolution wavelet decomposition to generate time–frequency patches which are then embedded and mixed via lightweight MLP modules, efficiently capturing both local and global patterns for long-term time series forecasting (Murad et al., 2025).

- **TimeLLM:** This method is a reprogramming framework to repurpose LLMs for general time series forecasting with the backbone language models kept intact (Jin et al., 2024).

### A.3 EXPERIMENTS

### A.3.1 OVERALL PERFORMANCE COMPARISON

Table 2: Overall Performance Comparison.

| Methods | Dataset | ETTm2→Traffic | | | Weather→Electricity | | | Electricity→ETTm2 | | | Traffic→Weather | | | ETTh2→Weather | | | ETTm1→ETTh1 | | |
|---------|---------|-------|-------|-------|-------|-------|-------|-------|-------|-------|-------|-------|-------|-------|-------|-------|-------|-------|-------|
| | PL | 96 | 192 | 336 | 96 | 192 | 336 | 96 | 192 | 336 | 96 | 192 | 336 | 96 | 192 | 336 | 96 | 192 | 336 |
| OFA | MSE | 0.455 | 0.469 | 0.503 | 0.171 | 0.197 | 0.271 | 0.185 | 0.242 | 0.293 | 0.239 | 0.254 | 0.299 | 0.221 | 0.288 | 0.299 | 0.482 | 0.502 | 0.498 |
| | MAE | 0.336 | 0.344 | 0.368 | 0.279 | 0.293 | 0.313 | 0.270 | 0.306 | 0.337 | 0.295 | 0.297 | 0.328 | 0.277 | 0.325 | 0.322 | 0.470 | 0.486 | 0.484 |
| TimePD | MSE | **0.453** | **0.469** | **0.496** | **0.170** | **0.189** | **0.211** | **0.180** | **0.229** | **0.281** | **0.229** | **0.236** | **0.271** | **0.168** | **0.287** | **0.271** | **0.456** | **0.474** | **0.441** |
| | MAE | **0.327** | **0.341** | **0.362** | **0.275** | **0.293** | **0.312** | **0.267** | **0.304** | **0.335** | **0.285** | **0.285** | **0.307** | **0.226** | **0.321** | **0.308** | **0.454** | **0.464** | **0.453** |

In this section, we present the experimental results that compare TimePD with OFA using other datasets (except ETTh1) as source data. The experimental results show that our method can also achieve better prediction results when other datasets are used as source data.

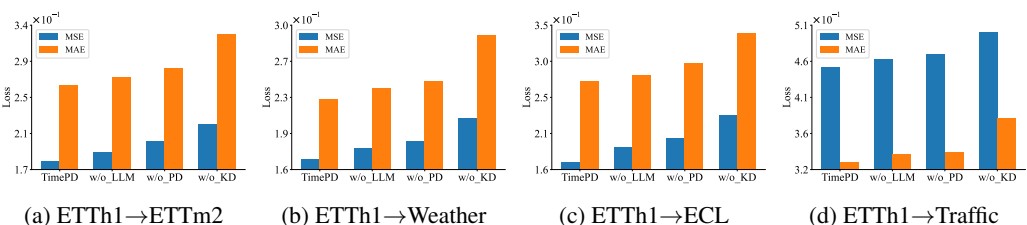

Figure 9: Performance of TimePD and Its Variants

### A.3.2 ABLATION STUDY

Here, we present the experimental results on the remaining datasets, as shown in Figure 9.

### A.3.3 EFFECT OF THE SIZE OF TARGET DOMAIN DATASET

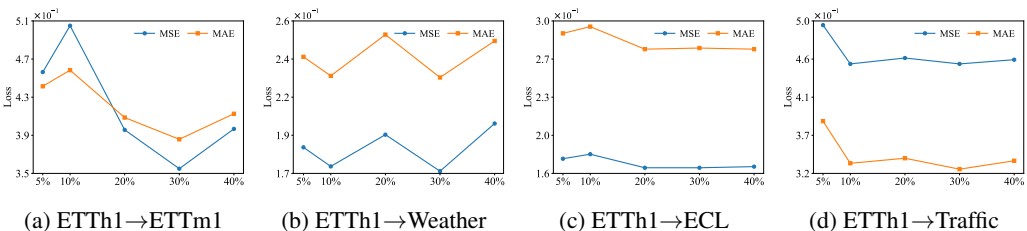

Figure 10: Effect of the Size of Target domain Dataset

Figure 10 shows experimental results on other datasets. In the main text, we mentioned that the degree of improvement varies significantly across datasets, and in some cases, additional data even leads to performance degradation. For example, the ETTh1 → ECL task shows very little variation across all data proportions, indicating that the model achieves near-optimal performance even with as little as 5% of target data. This insensitivity implies that the model transfers well to the ECL domain with minimal adaptation. Interestingly, ETTh1 → ETTm1 and ETTh1 → weather present non-monotonic trends. For example, in the ETTm1 task, performance initially worsens from 5% to 10% and then improves, while for weather, fluctuations occur throughout. This behavior may result from domain complexity, data noise, or overfitting due to insufficient generalization. For the ETTh1 → traffic task, model performance fluctuates within a narrow range across all data proportions. The lack of substantial improvement suggests that the model might have already captured the essential patterns with a small amount of target data, and further data adds limited value.

### A.3.4 EFFECT OF THE LEARNING RATE

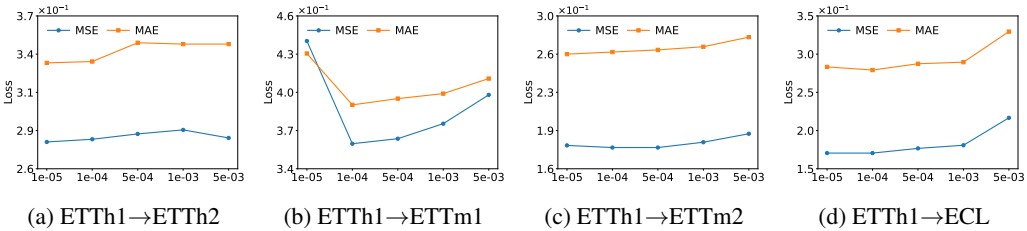

Figure 11: Effect of the learning rate

We further study the sensitivity of our model to the learning rate, as shown in Figure 11. Across all transfer settings, extremely small or large values lead to performance degradation, while moderate values (e.g., 1e-4) yield consistently better results. In particular, ETTh1→ETTm1 exhibits a sharp loss increase when deviating from this range, indicating that the model is relatively sensitive to the

learning rate in certain domains. Overall, these results suggest that our method remains robust within a reasonable range but requires careful tuning to achieve optimal performance.

### A.3.5 PSEUDO TRAINING CODE OF TIMEPD

---

**Algorithm 1:** The TimePD Framework

---

**Input:** Source model $\theta_s$, Pre-trained large language model $\theta_{ts}$, Source dataset $X_S$, Target dataset $X_T$, Denoising strength $\alpha$, Loss weights $\lambda_1, \lambda_2$, Iterations $M$
**Output:** Adapted target model $\theta_t$
**Initialization:** Train $\theta_s$ on source dataset $X_S$ and set target model $\theta_t \leftarrow \theta_s$.
**for** $m = 1$ *to* $M$ **do**

    Sample a mini-batch $X_T^b$ from $X_T$.
    Obtain source prediction $z_s$ by forwarding $X_T^b$ through $\theta_s$ (frozen).
    Obtain target prediction $z_t$ by forwarding $X_T^b$ through $\theta_t$.
    Obtain proxy forecast $z_{proxy}$ by forwarding $X_T^b$ through $\theta_{ts}$(frozen).
    Extract invariant trend and seasonal features $h_{trend}, h_{season}$ via invariant feature learning.
    Compute invariance regularizers $L, L_{inv}, L_{pred}, L_{rep}, L_{grad}$ at representation and gradient levels (Eq. (4), Eq. (10), Eq. (15), Eq. (16), Eq. (18)).
    Apply proxy denoising to correct proxy forecasts of $X_T^b$ (Eq. (19)).
    $z_{denoised} \leftarrow z_{proxy} - \alpha(z_s - z_t)$.
    Apply knowledge distillation between corrected proxy $z_{denoised}$ and target outputs $z_t$ (Eq. (20)).
    $L_{mkd} \leftarrow MSE(z_{denoised}, z_t)$.
    Compute the overall objective $L_{\text{all}}$ and update $\theta_t$ by minimizing $L_{\text{all}}$ (Eq. (21)).

**return** *Adapted target model $\theta_t$*

---

We show the training process of TimePD in Algorithm 1. With the optimization objective proposed in Eq. (21), we can effectively train and optimize the model.

### A.4 THE USE OF LARGE LANGUAGE MODELS (LLMS)

LLMs are used in this work solely for auxiliary purposes. Specifically, they assisted in improving the accuracy of writing by identifying and correcting grammatical issues. All research ideas, methodological developments, experiments, and the main body of the manuscript are independently conceived, conducted, and written by the authors.

