# OpenReview forum: "Deciphering Invariant Feature Decoupling in Source-free Time Series Forecasting with Proxy Denoising"
_ICLR.cc/2026/Conference — ICLR 2026 Conference Withdrawn Submission_

### Official Review · Reviewer_HYWG · 2025-10-20

**Soundness:** 3
**Presentation:** 3
**Contribution:** 3
**Rating:** 6
**Confidence:** 3

**Summary:**

This paper proposes TimePD, a source-free domain adaptation framework for time series forecasting. Unlike traditional transfer learning approaches that require both source and target data, TimePD adapts a pretrained source model to a target domain without direct access to the source data.

The framework combines three main ideas:
1. Invariant Disentangled Feature Learning (IDFL): a dual-branch seasonal/trend decomposition with both representation- and gradient-level explicit invariance, enhanced with frequency-consistent supervision via Fourier module.
2. Proxy Denoising (PD): treats an LLM as a noisy forecaster and dynamically corrects its systematic bias using source–target model consensus.
3. Knowledge Distillation (KD): transfers denoised, LLM-guided knowledge into a lightweight target model.

**Strengths:**

- **Practical relevance**: Source-free adaptation setting is aligned with real-world deployment constraints (e.g., data privacy, federated setups).
- **Conceptual novelty**: Combination of disentangled trend/seasonal invariance, LLM-guided proxy denoising, and bidirectional knowledge transfer is creative and well motivated.
- **Interpretability and visualization**: Decomposition-based structure provides interpretable insights (trend and seasonal components).

**Weaknesses:**

1. **Ablation insufficiency**: While IDFL is one of the core contributions, there is no w/o IDFL experiment to show its effectiveness.
2. **KD loss definition**: Because the KD correction term includes target model $\theta_t$ inside the teacher signal, the loss may partially self-cancel or destabilize learning.
3. **LLM proxy validity**: If the chosen LLM backbone has already seen the target data during pretraining, its role as a _generalizable proxy_ becomes questionable — it may behave as a well-trained teacher rather than an unbiased generalizer.
4. **Computational overhead and scalability**: While it is understandable that training an LLM requires substantial computation, the gradient-level alignment also involves repeated $\partial L/\partial\theta$ computations, which are memory- and compute-intensive.
5. **Overparameterization and sensitivity**: Since IDFL involves multiple hyperparameters (e.g., $k_\text{trend}$, $\alpha$-percentile threshold, loss weights), robustness or sensitivity analyses should be included to better support the model’s stability.

**Questions:**

1. Could you please include an ablation study with `w/o_IDFL` and, if possible, further decompose it into subcomponents (e.g., `w/o_rep-align`, `w/o_grad-align`, etc.) to better clarify each part’s contribution?
2. Could you elaborate on how training instability is avoided for the KD term in Eq. (20)? Is there any special treatment (e.g., stop-gradient) applied?
3. Could you provide more details on how the OFA model is utilized? Was it pretrained on any domains similar to your target datasets? How do you ensure that it functions as a general proxy rather than a domain-specific teacher?
4. Would it be possible to report the computational cost (e.g., GPU hours, memory usage) and discuss the scalability of your method with respect to sequence length and data dimensionality?
5. Have you considered adding robustness or sensitivity analyses for key hyperparameters to better assess the model’s stability?

---

### Official Review · Reviewer_6fd8 · 2025-10-28

**Soundness:** 1
**Presentation:** 2
**Contribution:** 1
**Rating:** 2
**Confidence:** 3

**Summary:**

This paper introduces TimePD, a framework for source-free domain adaptation (SFDA) in time series forecasting (TSF). The problem setting involves adapting a pre-trained model from a source domain with abundant data to a target domain with sparse data, without access to the source data. The proposed method leverages large language models (LLMs) as a proxy forecaster, incorporating three main components: (1) Invariant Disentangled Feature Learning (IDFL), which decomposes time series into seasonal and trend components and enforces invariance at representation and gradient levels; (2) Proxy Denoising (PD), which corrects LLM hallucinations by calibrating predictions using consensus between source and target models; and (3) Knowledge Distillation (KD), which aligns the target model's outputs with denoised LLM predictions. Experiments on real-world datasets (e.g., weather, electricity, traffic, ETT) claim superior performance over baselines.

**Strengths:**

1. The paper addresses an underexplored problem: source-free TSF, which is relevant for privacy-constrained scenarios in IoT and sensor data applications.

2. The framework includes some intuitive elements, such as season-trend decomposition and bidirectional knowledge transfer, which could inspire future work in cross-domain TSF.

**Weaknesses:**

1. Methodological Concerns and Lack of Novelty: The core of IDFL relies on gradient-based masking to enforce invariance (e.g., computing gradient differences in Eq. 6-8 and aligning gradients in Eq. 18). While this aims to disentangle seasonal and trend components, the approach feels somewhat arbitrary and lacks strong theoretical grounding. For instance, ranking absolute gradient differences and applying a percentile-based binary mask (Eq. 7) appears ad hoc—why this specific thresholding mechanism over more established invariance techniques (e.g., from causal representation learning or domain generalization literature)? It risks overfitting to spurious correlations rather than capturing true invariants, especially in sparse target data where gradients may be noisy. Furthermore, the overall architecture seems like a patchwork of existing ideas: season-trend decomposition via moving averages (Eq. 1) is standard in TSF; the forecaster is directly adopted from TSFE (Miao et al., 2024); Fourier-based supervision (Eq. 11-15) echoes frequency-domain methods in recent TSF works; and gradient alignment draws from Chen et al. (2025b). The PD module, while novel in intent, simplifies hallucination correction to a linear subtraction (Eq. 19) based on source-target residuals, which may not robustly handle diverse LLM biases. KD (Eq. 20) is also straightforward MSE alignment, adding little beyond standard distillation. Collectively, this gives the impression of incremental tweaks and component assembly rather than a cohesive, innovative contribution to the field. The multi-part loss (combining $L, L_{inv}, L_{pred}, L_{rep}, L_{grad}, L_{kd}$) exacerbates this, as it could lead to hyperparameter sensitivity without clear ablation on how each term uniquely drives performance.

2. Experimental Design and Significance of the Problem: The experiments appear limited in scope and fail to demonstrate the problem's broader relevance. The setup involves training on one domain and adapting to sparse target domains, but this cross-domain transfer is not compared to zero-shot or few-shot capabilities of recent foundation TSF models like MOIRAI, TimerXL or Time-Moe. Such comparisons are crucial, as these models often achieve strong generalization on sparse data via pre-training on massive corpora, potentially rendering SFDA redundant in many cases. Ablations seem insufficient to isolate contributions, and the lack of sensitivity analysis raises questions about robustness. Overall, the problem of SF-TSF, while conceptually interesting, is not convincingly motivated as a critical gap—especially when privacy concerns could be addressed via federated learning or synthetic data generation in TSF.

3. Minor Issues: The writing has some inconsistencies, and figures (e.g., Fig. 3) could better illustrate gradient flows. The assumption that source and target share latent patterns is not empirically validated in extreme domain shifts.

**Questions:**

1. Can you provide justification for the gradient-based masking over alternatives like mutual information minimization or adversarial invariance?

2. Why no comparisons to foundation TSF models in zero/few-shot settings? How does TimePD perform against them on the same sparse targets?

3. How sensitive is performance to hyperparameters like $\sigma, k_{trend}$, or the percentile in masking?

---

### Official Review · Reviewer_XRmj · 2025-10-31

**Soundness:** 2
**Presentation:** 3
**Contribution:** 2
**Rating:** 2
**Confidence:** 5

**Summary:**

This paper considers a task of source-free time-series forecasting and proposes TimePD, which (1) performs dual-branch invariant disentanglement (season and trend) with representation- and gradient-level alignment, (2) uses an LLM as a proxy forecaster and applies proxy denoising to correct its systematic bias, and (3) distills the denoised proxy into a lightweight target model. Experiments on widely-used time series forecasting datasets demonstrate the effectiveness of TimePD.

**Strengths:**

**S1.** This paper is well-organized and easy to follow.

**S2.** The problem of source-free domain adaptation for time series forecasting is make sense.

**S3.** The workflow of Invariant Disentangled Feature Learning is reasonable.

**Weaknesses:**

**W1.** Representation-level masking selects the $\alpha$-percentile of gradient-difference entries to zero (Eq. 6–8). The paper does not specify the value of $\alpha$, how it is tuned, or the compute overhead of repeatedly computing per-branch gradients for masking within each iteration. In addition, this step is discrete/rank-based and could be unstable.

**W2.** The DFT module partitions frequencies with a fixed cut-off index $k_{\text{cut}}$, but the details about it are missing.

**W3.** In SF experiments, the paper does not clarify whether the baselines are constrained to the same label availability and supervision budget as TimePD.

**W4.** The final optimization objective (Eq. 21) is inevitably complex because it involves up to five constraint terms controlled by five hyperparameters. This means that TimePD is difficult to train, which is a significant concern for me.

**W5.** Based on W4, the paper lacks of no hyper-parameter analysis for $\alpha, \lambda_{\text{inv}},\lambda_{\text{pred}},\lambda_{\text{rep}},\lambda_{\text{grad}},\lambda_{\text{kd}}$.

**W6.** This paper employs TSLib Benchmark as its evaluation dataset. However, the performance of these datasets has reached saturation and exhibits certain irregular processing patterns (despite their widespread adoption), which significantly limits the validity of the claims made in TimePD.

**Questions:**

Please see Weaknesses.

Additional questions:

**Q1.** How is $k_{\text{cut}}$ determined for each dataset/window length?

---

### Official Review · Reviewer_PTM5 · 2025-11-01

**Soundness:** 3
**Presentation:** 3
**Contribution:** 2
**Rating:** 4
**Confidence:** 4

**Summary:**

The paper studies source-free time-series forecasting, where only a pretrained source model is available and the target domain offers limited data. It introduces TimePD, a three-part framework: (1) an invariant, disentangled representation that explicitly separates trend and seasonal components and enforces consistency across domains, (2) a proxy denoising step that treats an LLM forecaster as a noisy teacher and calibrates its outputs using the agreement between source and target predictors, and (3) knowledge distillation that transfers the denoised, high-level temporal patterns into a lightweight target model. The experiments across multiple cross-domain shows consistent improvements over strong LLM-based baselines on low-resource domain transfer.

**Strengths:**

- The authors proposed the first source-free time series forecasting methodology.
- The empirical results show consistent improvements across diverse transfer pairs and horizons
- Ablations indicate that each component contributes meaningfully.

**Weaknesses:**

- The method may rely on the source and target exhibiting similar seasonal and trend structure, yet the failure modes of the decomposition are insufficiently analyzed.
- The method’s behavior on datasets with weak or ambiguous periodic structure is underexplored, leaving the decomposition’s robustness uncertain.
- The absence of a backbone-only baseline obscures how much improvement stems from the proposed framework versus the underlying forecaster.

**Questions:**

- The current study fixes the target-domain supervision at 30%, but only analyzes TimePD’s ablations. To assess sample efficiency and robustness, please compare other baselines across multiple target-data fractions (e.g., 5/10/20/30/50/100%).
- Figures 8 (a) and (b) report season and trend comparisons on the ETT dataset. Could you specify which subsets were used (ETTh1/ETTh2/ETTm1/ETTm2)?
- It would also be helpful to include an invariant feature visualization on datasets with weak or ambiguous seasonality and trend.
- Would you provide a TSFE-only ablation or baseline table to clarify how much each module contributes beyond the underlying forecaster?
- Could you quantify the performance gap between your approach and baseline models trained from scratch, using identical data splits and budgets?

---

### Note · Authors · 2025-12-05

I have read and agree with the venue's withdrawal policy on behalf of myself and my co-authors.